# What is Adversarial Training for Diffusion Models?

## Abstract

We answer the question in the title showing that adversarial training (AT) for diffusion models (DMs) is inherently different from classifiers. Whereas for the latter it is related to *invariance* of the output given input from a fixed class, AT for DMs requires *equivariance* to make the diffusion process still land in the data distribution. For the first time, we define AT as a means to enforce smoothness in the diffusion flow to make it more resistant to outliers or corrupted datasets. Unlike prior art, ours does not require any particular assumption on the noise model and our new training scheme can be implemented on top of the diffusion noise, using additional random noise—similar to randomized smoothing—or adversarial noise—akin to adversarial training. Our method unlocks capabilities such as intrinsically handling noisy data, dealing with extreme variability such as outliers, preventing memorization, and, obviously, improving robustness and security. We rigorously evaluate our approach with proof-of-concept datasets with *known* distributions in low- and high-dimensional space, thereby taking perfect measure of errors; we further evaluate on standard benchmarks such as CIFAR-10, recovering the underlying distribution in presence of strong noise or corrupted data.

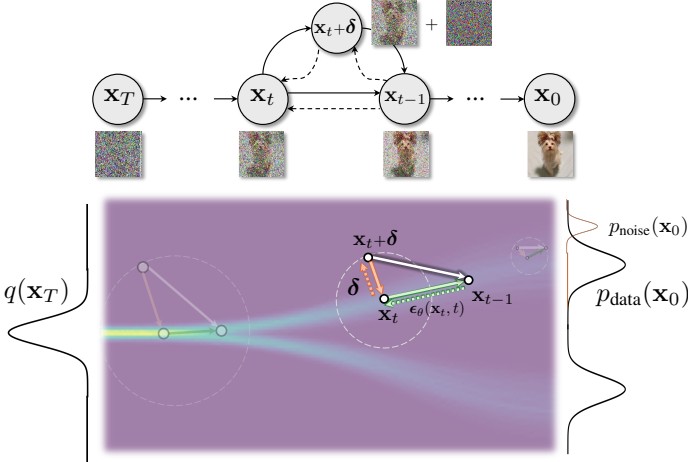

Figure 1. **Inducing smoothness into diffusion trajectories.** Instead of training directly the denoising network to follow the score function *i.e.*, $\mathbf{x}_t \mapsto \mathbf{x}_{t-1}$ using just $\boldsymbol{\epsilon}_\theta(\mathbf{x}_t, t)$, we locally perturb the data point as $\mathbf{x}_t + \boldsymbol{\delta}$ inside a $\ell_p$ ball centered on $\mathbf{x}_t$ and impose equivariance to let the model reach the same point yet passing through the local perturbation: $\mathbf{x}_t + \boldsymbol{\delta} \mapsto \boldsymbol{\epsilon}_\theta(\mathbf{x}_t, t) + \boldsymbol{\delta} \triangleq \mathbf{x}_{t-1}$. This is equivalent to adding an intermediate step between the Markov Chain that functions as an additional denoising step during training in case the network is misled by outliers or noise in the dataset—$p_{\text{noise}}(\mathbf{x}_0)$—not proper of $p_{\text{data}}(\mathbf{x}_0)$. The local noising step can be implemented as adversarial (Goodfellow et al., 2015) or as random, akin to randomized smoothing, Cohen et al. (2019). The perturbation is adaptive so that is large in the noise phase and shrinks in the content phase. ⟶ indicates the forward process; - - -⟶ the reverse process.

## 1. Introduction

When we train Diffusion Models (DMs) going large-scale, noise in the data is inevitable. Training generative modeling on a massive amount of data is key for the current success of AI, alas, data-level noise can arise in different forms such as inlier noise, minor perturbations to data points that remain within the expected distribution, or outlier noise, data points that deviate significantly from the others, going out from the

expected distribution. We may also have missing data or corrupted data where values may be affected by Gaussian noise; finally, we may have "adversarial" noise such as those found in poisoning attacks (Tian et al., 2022). Although there have been attempts to train in noisy settings with some remarkable results such as Daras et al. (2024c;d;a), it is desirable to have a generic method that makes less assumptions on the type of noise. In fact, Daras et al. (2024c) requires knowing the exact variance of the added Gaussian noise, Daras et al. (2024d) handles only missing data and (Daras et al., 2024a) needs to know which samples are noisy and which are clean. Another open problem with DMs is the fact that despite their remarkable generation performance (Dhariwal & Nichol, 2021), researchers in AI Safety and red-teaming

[1]Anonymous Institution, Anonymous City, Anonymous Region, Anonymous Country. Correspondence to: Anonymous Author <anon.email@domain.com>.

Preliminary work. Under review by the International Conference on Machine Learning (ICML). Do not distribute.

have demonstrated that they memorize part of the training data (Jagielski et al., 2023; Somepalli et al., 2023; Carlini et al., 2023). So DMs have solved GAN stability problems, such as mode collapse, yet they have introduced the issue of memorization and leakage of information. Indeed, while DMs may better cover the modes of the underlying data distribution (Zhong et al., 2019) in the extreme case of over-parametrized models, this may lead to the aforementioned memorization. While it is important to cover all the modes, we may want to discard minor modes, latent factors that are not proper of the input data manifold or spurious correlations. To strike a better trade-off between mode coverage and memorization, we propose a method to smooth the trajectory space of DMs as depicted in Fig. 1. Leveraging on the limits of prior art, we make the following contributions:

◇ Despite a few papers applied AT to DMs (Yang et al., 2024; Sauer et al., 2024), so far no one has defined what AT is for DMs at the fundamental level and shed light on when is convenient to apply it, i.e. denoising the data distribution. Though previous work made progress in handling noise with DMs (Daras et al., 2024d;c;a), we are the first to reconnect AT as a denoising method.

◇ Inspired by Zhang et al. (2019) for classifiers, we define a proper algorithm to apply AT to DMs, showing that it is inherently different from classification: while a categorization problem requires developing *invariance*, for a score-matching problem, there is the need for *equivariance* as summarized in our key finding in Eq. (10).

◇ We empirically demonstrate the flexibility of our method in handling noisy data, dealing with extreme variability such as outliers, preventing memorization, and improving robustness and security. We do so by providing experiments in low-dimensional settings (3D) with known distributions and also in high-dimension still with known distributions. Following prior art (Daras et al., 2024c;a) we use Gaussian noise to perturb the data. We evaluate on a real dataset such as CIFAR-10, scoring good metrics even in the presence of extreme clutter.

## 2. AT smooths the Diffusion Flow

### 2.1. Preliminaries

Diffusion Models (DMs) (Ho et al., 2020) aim to learn a data distribution, $p_{\text{data}}(\mathbf{x})$ by "encoding data" using a fixed noising procedure that maps data to $\mathcal{N}(\mathbf{0}, \mathbf{I})$ using a Markov chain $q(\mathbf{x}_T, \dots, \mathbf{x}_1 | \mathbf{x}_0) = \prod_{t=1}^{T} q(\mathbf{x}_t | \mathbf{x}_{t-1})$, where, given a noisy input $\mathbf{x}_{t-1}$, the distribution of the next state is:

$$q(\mathbf{x}_t | \mathbf{x}_{t-1}) = \mathcal{N}\big(\mathbf{x}_t; \sqrt{1 - \sigma(t)} \mathbf{x}_{t-1}, \sigma(t) \mathbf{I}\big), \quad (1)$$

and $\sigma(t)$ is the noise scheduler: a monotonically decreasing time-varying function chosen such that $\sigma(0) = \sigma_{\min}$ and

$\sigma(T) = \sigma_{\max}$ and often $0 < \sigma_{\min} < \sigma_{\max} < 1$. Generation is attained with a learnable "decoding step" that reverts back data from noise estimating $p(\mathbf{x}_{t-1} | \mathbf{x}_t)$. If the noise scheduler is chosen carefully to take small noising steps, then the approximation $q(\mathbf{x}_T | \mathbf{x}_0) \approx \mathcal{N}(\mathbf{0}, \mathbf{I})$ and the following equation hold:

$$q(\mathbf{x}_t | \mathbf{x}_0) = \mathcal{N}\big(\mathbf{x}_t; \sqrt{\alpha_t} \mathbf{x}_{t-1}, (1 - \alpha_t)\mathbf{I}\big), \ \alpha_t \doteq \prod_{s=1}^{t} 1 - \sigma(t)$$

This means we can encode directly from $\mathbf{x}_0 \mapsto \mathbf{x}_t$ as:

$$\mathbf{x}_t = \sqrt{\alpha_t} \mathbf{x}_0 + \sqrt{1 - \alpha_t}\, \boldsymbol{\epsilon} \ \text{ where } \ \boldsymbol{\epsilon} \sim \mathcal{N}(\mathbf{0}, \mathbf{I}). \quad (2)$$

The process is analyzed in Song et al. (2021b) as denoising score matching, following the Stochastic Differential Equation (SDE):

$$\mathbf{x}_t = \mathbf{f}(\mathbf{x}_t, t)t + g(t)\mathbf{w}_t \quad (3)$$

where $\mathbf{w}$ is the standard Wiener process, $\mathbf{f}(\cdot, t) : \mathbb{R}^d \to \mathbb{R}^d$ is the drift coefficient, and $g(\cdot) : \mathbb{R} \to \mathbb{R}$ is the diffusion coefficient. Generation is performed by solving the probability flow ODE (PF-ODE), from $t = T$ to $0$ and starting from $\mathbf{x}_T \sim \mathcal{N}(0, \sigma_{\max}^2 I)$, whose solution is learned from the DM. For a given $\mathbf{x}_0$, the simplified version of training objective $\mathcal{L}_{\text{DM}}$ reported in Ho et al. (2020) is thus defined as:

$$\mathcal{L}_{\text{DM}} = \mathbb{E}_{\substack{\boldsymbol{\epsilon} \sim \mathcal{N}(\mathbf{0}, \mathbf{I}) \\ t \sim \mathcal{U}(\mathbf{0}, \mathbf{I})}} \left[ \big\| \boldsymbol{\epsilon} - \boldsymbol{\epsilon}_\theta \big(\mathbf{x}_t(\mathbf{x}_0, \boldsymbol{\epsilon}), t\big) \big\|_2^2 \right] \quad (4)$$

whose objective is to infer the noise $\boldsymbol{\epsilon}$ applied to the initial image, ensuring that the starting point $\mathbf{x}_0$ is correctly reconstructed, enabling the model—the denoising network $\boldsymbol{\epsilon}_\theta$—to correctly generate in-distribution data during inference. For inference we solve the SDE using $\boldsymbol{\epsilon}_\theta$ and the recurrency:

$$\mathbf{x}_{t-1}(\theta) = \frac{1}{\sqrt{1 - \sigma(t)}} \left( \mathbf{x}_t(\theta) - \frac{\sigma(t)}{\sqrt{1 - \alpha_t}} \boldsymbol{\epsilon}_\theta \big(\mathbf{x}_t(\theta), t\big) \right) +$$
$$+ \sigma(t)\mathbf{z} \qquad \mathbf{z} \sim \mathcal{N}(\mathbf{0}, \mathbf{I}) \ \text{ and } \ \forall t \in [T, \dots, 0].$$
$$(5)$$

### 2.2. Motivation, "in vitro" experiments, and noise types

**Motivation and overview.** Adversarial samples for classifiers are the cornerstone idea towards robust models (Goodfellow et al., 2015; Madry et al., 2018). To ensure the invariance of classifier response to future adversarial perturbations, AT is designed to maintain consistency of the model's output. Unlike classifiers, training DMs involves a regression task, and the AT problem must be formulated differently. In our work, we propose the first principled approach to adversarially train diffusion models. As shown in Fig. 2(a) our methods can estimate the underlying distribution even in presence of strong inliers noise or uniform outliers; moreover, considering Fig. 2(b), it can smooth the score fields of DMs leading to more stable fields.

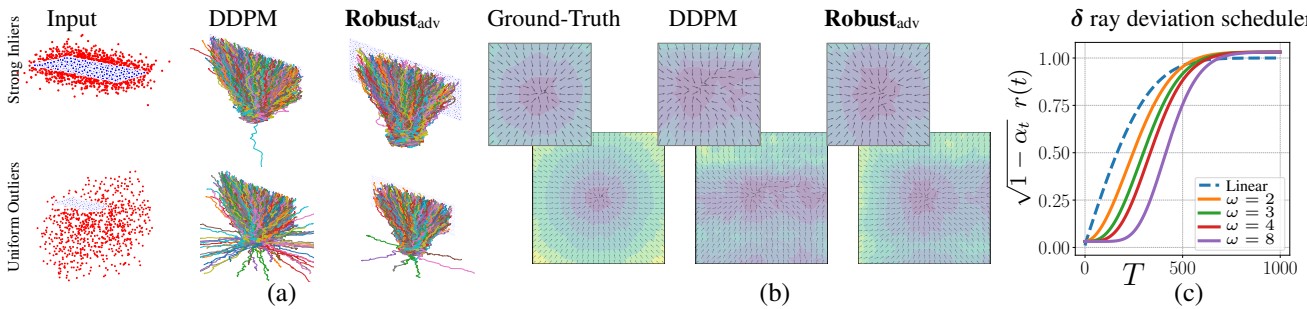

*Figure 2.* (a) Handling different types of noise. While the baseline method DDPM (Ho et al., 2020) struggles to handle distributions where the true one is embedded with either strong inliers noise *(top)* or uniform outliers *(bottom)*, ours is more robust. (b) Score vector fields: versors represent the score field, where their magnitude is represented as colormap, ██ less ██ more intense. *(left)* Ground-truth *(middle)* Baseline DDPM (Ho et al., 2020); *(right)* Our **Robust**adv. AT induces more smooth and consistent score, better matching the shape of the input distribution. As side effects: it shrinks the variability of the data distribution and induces more intense fields. (c) Adversarial perturbation ray. Different curves show different trends varying the $\omega$ parameter. In DDPM, $\sqrt{1-\alpha_t}\ r(t)$: we use a similar scheduler as $r(t)$ but the exponent $\omega$ shrinks the content phase and pushes down the slope of the curve to reduce it.

**"In vitro" Experimental Setup.** We experiment on synthetic 3D datasets where we have the possibility to go from "linear" and unimodal datasets to more complex multi-modal. In `oblique-plane` we assume $p_{\text{data}}$ lives approximately on a 2D subspace with equation $x + y + z = 30$ embedded in 3D. In `3-gaussians`, we address multi-modal distributions such as Mixture of Gaussian $\frac{1}{3}\mathcal{N}([10, 10, 10], \sigma) + \frac{1}{3}\mathcal{N}([20, 20, 20], \sigma) + \frac{1}{3}\mathcal{N}([10, 30, 30], \sigma)$ with $\sigma = 0.25$. We also move to high-dimensional space $\mathbf{x} \in \mathbb{R}^{32 \times 32 \times 3}$ yet still in controlled settings by using a simple dataset of images of butterflies, "the Smithsonian Butterflies" [1]. It consists of aligned `butterflies` which we resize to $32 \times 32$ and then flatten. We then linearize the data by fitting a subspace of 25 dimensions embedded in a 3072-D space, retaining 70% of the variance using principal components as $\mathbf{x}' = \boldsymbol{\mu} + \sum_i \lambda_i \boldsymbol{\alpha}_i \mathbf{U}_i$ where the stochasticity comes from $\boldsymbol{\alpha} \sim \mathcal{N}(0; \sigma)$, then $\boldsymbol{\mu} \in \mathbb{R}^{3072}$ and $\mathbf{U} \in \mathbb{R}^{25 \times 3072}$ are the mean and the principal components of the datasets, and $\lambda_i$ is the singular value associated to each component. The new linearized data is visually very similar to the real data, thus we throw away the real data and proceed to train DMs to fit $\{\mathbf{x}'\}_{i=1}^N$. This allows us to have a perfect measure of errors of how much we arrive close to the distribution. In this case, for example, we avoid measuring proxy metrics such as FID, instead measuring the reconstruction error between the subspace and the image generated by the diffusion model as $\rho = \|\mathbf{x}_0(\theta) - \mathbf{U}\mathbf{U}^\top \mathbf{x}_0(\theta)\|$ where $\mathbf{x}_0(\theta)$ is the generation using Eq. (5). In addition, we also measure the Peak Signal-to-Noise Ratio (PSNR).

**Noise model tested.** We experiment with different noise models: for 3D point clouds we add inlier noise by increasing the $\sigma$ of Gaussians or increasing $\boldsymbol{\alpha}$ as $\boldsymbol{\alpha} \sim \mathcal{N}(0; \sigma)$ in case of subspace $\boldsymbol{\mu} + \sum_i \lambda_i \boldsymbol{\alpha}_i \mathbf{U}_i$. We also experiment with outlier noise by adding strong noise in the ambient space:

---
[1]Dataset available on Hugging Face

for 3D point clouds we embed the original point cloud with dense, grid-like uniform noise; for `butterflies` in high-dimension, we simply add Gaussian noise after linearizing as $\mathbf{x}' + \mathbf{z}$ where $\mathbf{z} \sim \mathcal{N}(\mathbf{0}, \sigma\mathbf{I})$. Fig. 2(a) and Fig. 3(a) show the noise types and the datasets.

### 2.3. Injecting Adversarial Noise in the Diffusion Flow

**Injecting additional noise in the diffusion flow.** Given that DM encoding process already perturbs the data with Gaussian noise, it is not trivial to add another perturbation. After extensive tests and research, we found out that it is necessary to craft the $\boldsymbol{\delta}$ perturbation following these requirements: i) the ray of the perturbation, i.e. $r(t) = \|\boldsymbol{\delta}(t)\|_p$ has to be time-dependent following the noise scheduler $\sigma(t)$; ii) recalling the diffusion phases as defined in Choi et al. (2022), the ray cannot be large in the content phase, otherwise the approach may merge two different modes of the data distribution yielding over smoothing; iii) when $t \to T$, thus we are close to pure noise, $r(t)$ can have a high value with a maximum of 1 to maintain the assumption mentioned in Section 2.1; iv) finally, for $t \to 0$, $r(t)$ also have to tend to zero yet keeping a constant bias $\gamma$ at the end; if we do not do so, the approach may under smooth the data and the denoising will not occur. According to this, we modify Eq. (2) as:

$$\mathbf{x}_t = \sqrt{\alpha_t}\mathbf{x}_0 + \sqrt{1-\alpha_t}\,(\boldsymbol{\epsilon}+\boldsymbol{\delta}) \ \text{ where } \ \boldsymbol{\epsilon} \sim \mathcal{N}(\mathbf{0}, \mathbf{I}), \ (6)$$

and $\boldsymbol{\delta}$ is sampled independently for each dimension as:

$$\boldsymbol{\delta} \sim \mathcal{U}\big(-r_\beta(t), r_\beta(t)\big) \text{ and } r_\beta(t) \doteq \frac{(\sqrt{1-\alpha_t})^\omega + \gamma \cdot \beta}{\sqrt{1-\alpha_t}} \tag{7}$$

where $\omega \geq 1$ is an exponent to shrink the ray in the content phase and $\gamma$ is the bias to keep the ray at a minimum but not zero. The denominator in Eq. (7) is needed so that it

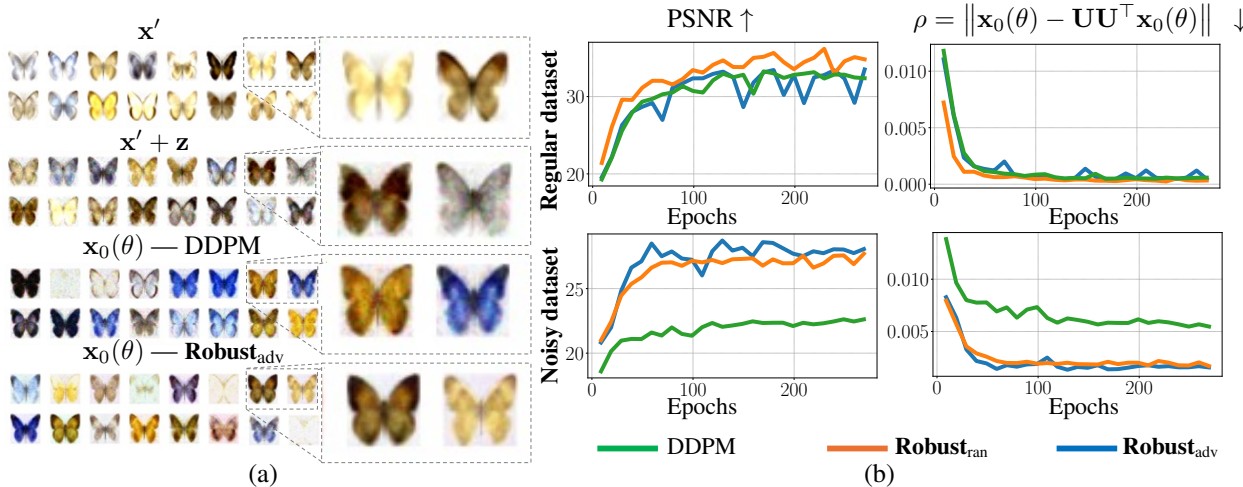

*Figure 3.* (a) Training on linearized `butterflies` dataset makes it possible to measure in closed form reconstruction error. From top to bottom, we show: the training data, the corrupted data, the generation results by DDPM and by **Robust**$_{\text{adv}}$. (b) Plot of metrics: first column PSNR while second column offers the closed-form reconstruction error. The first row displays regular, uncorrupted data, while the second row shows results with corrupted data where 90% of the data has been perturbed with Gaussian noise with $\sigma = 0.1$.

simplifies with $\sqrt{1 - \alpha_t}$ of Eq. (6). $\beta$ is a scalar to simply increase the bias randomly and by default is set to 1. Fig. 2(c) shows the adversarial perturbation ray in function of time compared to the normal DDPM scheduler.

**Two types of perturbations: random vs adversarial.** The additional noise we add to enforce smoothness can be of two types: random $\boldsymbol{\delta}_{\text{ran}}$, akin randomized smoothing (Cohen et al., 2019), or adversarial $\boldsymbol{\delta}_{\text{adv}}$, similar to AT (Goodfellow et al., 2015). **Random:** The random noise applied to enforce smoothness is defined as Eq. (7) yet sampling $\beta \sim \mathcal{U}[0.5, 2]$, a stochastic parameter included to randomize the ray so that the process is resilient to variability in the ray. Being defined in such a way, $\boldsymbol{\delta}_{\text{ran}}$ would be an uniform variable whose standard deviation is defined as $r_\beta(t)/\sqrt{3}$. **Adversarial:** The adversarial noise definition builds on top of the previous setting. The perturbation first is initialized randomly as $\boldsymbol{\delta}_{\text{ran}}$ and then updated similarly to Fast Gradient Sign Method (FGSM) with random start as in Kurakin et al. (2017) with $\ell_\infty$. We thus optimize $\boldsymbol{\delta}_{\text{adv}}$ as:

$$\boldsymbol{\delta}_{\text{adv}} \triangleq \underset{\|\boldsymbol{\delta}\|_\infty \leq r_\beta(t)}{\arg\min} \left\| \boldsymbol{\epsilon}_\theta \big( \mathbf{x}_t + \boldsymbol{\delta}, t \big) - \boldsymbol{\epsilon}_\theta \big( \mathbf{x}_t, t \big) \right\|_2^2 \quad (8)$$

that was implemented by taking a gradient step and projecting it back to the feasible set using Projected Gradient Descent (PGD):

$$\mathbf{x}_t^{\text{adv}} = \mathbb{P}_{r_\beta(t)} \left[ \mathbf{x}_t + \frac{r_\beta(t)}{\sqrt{3}} \operatorname{sign} \left( \nabla_{\mathbf{x}_t} \mathcal{L}_{\text{DM}} \big( \boldsymbol{\epsilon}, \boldsymbol{\epsilon}_\theta \big( \mathbf{x}_t(\mathbf{x}_0, \boldsymbol{\epsilon}), t \big) \big) \right) \right]$$

where $\mathbb{P}_{r_\beta(t)}$ projects onto the surface of $\mathbf{x}_t$'s neighbor $\ell_\infty$-ball and $r_\beta(t)/\sqrt{3}$ is the standard deviation of the attack.

### 2.4. Adversarial Training for Diffusion Models

Once we have defined to how inject the perturbation into the diffusion flow, we proceed to define the AT algorithm.

**Naïve invariance does not work.** We have found that simply applying AT loss as in classifiers:

$$\mathcal{L} = \arg\min_\theta \left\| \boldsymbol{\epsilon}_\theta \big( \mathbf{x}_t + \boldsymbol{\delta}, t \big) - \boldsymbol{\epsilon}_\theta \big( \mathbf{x}_t, t \big) \right\|_2^2 \quad (9)$$

does not work. Eq. (9) brings the DM to learn another distribution different than $p_{\text{data}}(\mathbf{x})$. Indeed, in this case, the model learns a slightly shifted data distribution, causing the generation of noisy data that was injected when adding the perturbation $\boldsymbol{\delta}$. This resulted in worse FID because the model was not able to perform denoising and the generated data contained the injected noise.

**Key change is equivariance.** We found out that the right way to apply AT to DMs is *equivariance*. The intuition is depicted in the introductory Fig. 1. Since, in the end, the forward process still needs to "land" in the data distribution despite the additional perturbations $\boldsymbol{\delta}$, the network must learn to denoise from $\mathbf{x}_t + \boldsymbol{\delta}$ in such a way that the denoised point always matches the previous one in the chain $\mathbf{x}_{t-1}$. This objective is reached by taking into account $\boldsymbol{\delta}$ in the AT loss as $\arg\min_\theta \left\| \boldsymbol{\epsilon}_\theta \big( \mathbf{x}_t + \boldsymbol{\delta}, t \big) - [\boldsymbol{\epsilon} + \boldsymbol{\delta}] \right\|_2^2$.

**Our Training.** Following the above setting, the proposed training formulation builds on top of that. The noisy sample $\mathbf{x}_t$ is defined as in Eq. (2), while the perturbed noisy sample $\mathbf{x}_t + \boldsymbol{\delta}$ is defined as in Eq. (6), considering either $\boldsymbol{\delta}_{\text{ran}}$ or $\boldsymbol{\delta}_{\text{adv}}$ as perturbation elements. We have the regular term—$\mathcal{L}_{\text{DM}}$ as in Eq. (4)—where the method teaches the network to flow towards the data distribution using $\boldsymbol{\epsilon}$. Yet,

---

**Algorithm 1** Adv. Training for Diffusion Models

---

**Input:** dataset $\mathcal{D}$, model parameter $\theta$, max timestep $T$, noise scheduler $\alpha_t$, reg. strength $\lambda$, ray scheduler $r_\beta(t)$
**repeat**
    Sample $\mathbf{x} \sim \mathcal{D}$, timestep $t \in \mathcal{U}(\{0, \ldots, T\})$,
    $\boldsymbol{\epsilon} \sim \mathcal{N}(\mathbf{0}, \mathbf{I})$, $\beta \sim \mathcal{U}[0.5, 2]$, $\boldsymbol{\delta} \sim \mathcal{U}[-r_\beta(t), r_\beta(t)]$
    $\mathbf{x}_t = \sqrt{\bar{\alpha}_t}\mathbf{x}_0 + \sqrt{1 - \bar{\alpha}_t}\boldsymbol{\epsilon}$    w/   Eq. (2)
    $\mathbf{x}_t^{\text{adv}} = \mathbf{x}_t + \sqrt{1 - \bar{\alpha}_t}(\boldsymbol{\delta} + \boldsymbol{\epsilon})$ w/ Eq. (6), $\boldsymbol{\delta}$ as in Eq. (7)
    Take a gradient step as $-\nabla_\theta \mathcal{L}_{\text{AT}}(\mathbf{x}_t, \mathbf{x}_t^{\text{adv}}, t, \boldsymbol{\epsilon})$ Eq. (10)
**until** convergence

---

we added our novel term $\mathcal{L}_{\text{reg}}$ that *enforces equivariance and smoothness around the regular trajectory of the DM.* Our final formulation is thus:

$$\mathcal{L}_{\text{AT}}(\mathbf{x}_t, \mathbf{x}_t^{\text{adv}}, t, \boldsymbol{\epsilon}) = \arg\min_\theta \underbrace{\left\|\boldsymbol{\epsilon}_\theta(\mathbf{x}_t, t) - \boldsymbol{\epsilon}\right\|_2^2}_{\mathcal{L}_{\text{DM}} \text{ to fit data distr.}} +$$

$$+ \underbrace{\lambda_t \left\|\boldsymbol{\epsilon}_\theta(\mathbf{x}_t^{\text{adv}}, t) - \boldsymbol{\epsilon}_\theta(\mathbf{x}_t, t) - \boldsymbol{\delta}\right\|_2^2}_{\mathcal{L}_{\text{reg}} \text{ to enforce smoothness}} \quad (10)$$

where $\lambda_t$ is a time-dependent hyper-parameter that defines the strength of the regularization. This parameter, $\lambda_t = \frac{\lambda \cdot \sqrt{3}}{\beta \cdot r(t)}$, depends on a constant $\lambda$ and is rescaled according to the norm of the perturbation through its standard deviation. Eq. (10) is intended to be applied directly to the DDPM framework which is based on a $\boldsymbol{\epsilon}$-prediction network. The steps of our method are detailed in Algorithm 1.

## 3. Experimental Results

We use a progression of datasets that go from extremely controlled to real data. We experiment on synthetic 3D datasets where we have the possibility to simulate "linear" and unimodal distributions to more complex multi-modal datasets. We present results "in vitro" to guide our analysis on low-dimensional and also high-dimensional datasets taking perfect measure of errors. We offer results on a real dataset such as CIFAR-10, quantitatively assessing quality and diversity, using established metrics such as IS (Salimans et al., 2016) and FID (Heusel et al., 2017) evaluated on $10,000$ images. Following prior art (Daras et al., 2024c;a), we experiment with Gaussian noise as the main source of corruption $p_{\text{noise}}(\mathbf{x})$ and only work in challenging settings, testing a percentage $p$ of corrupted data of $p = 90\%$ with two levels of $\sigma = \{0.1, 0.2\}$. When computing FID, we always test on the *clean dataset* despite training with noisy datasets. Unlike prior art, our method does not exploit which samples are clean and which noisy, nor has knowledge of the $\sigma$ applied for corruption. Our methods are indicated by **Robust** where "adv" in the suffix uses adversarial perturbation and "ran" uses random. Finally, we show additional experiments that support our claims on less memorization,

faster sampling, and robustness to attacks. **Hyperparameter Settings.** We set $\omega = 2$ and $\gamma$ was set to $8/255$. The strength of regularization $\lambda$ is set to $0.3$: we have experienced that if we raise $\lambda$ to $0.5$ we get an over-smoothing effect whereas too low values prevent denoising.

### 3.1. Evaluation using DDPM and DDIM

**Controlled Experiments.** Fig. 3(b) shows the results when we train on high dim. data that lives on a subspace. In case we train on the clean, regular dataset, the baseline and our Robust DMs perform similarly though **Robust**ran has slightly better PSNR. When we train on the noisy dataset, $\{\mathbf{x}' + \mathbf{z}\}_{i=1}^N$, then both Robust DMs offer superior performance (orange and blue curves) with wide gaps compared to the baseline (green curve) in both PSNR and reconstruction error. In this case the **Robust**adv appears to be better at unlearning the noise. The generation from the baseline DDPM often provides samples with saturated blue colors that are unlikely to be found in the training set while our method has better fidelity—see Fig. 3(a).

**Random or adversarial?** We can also reply this question: by ablating the use of $\boldsymbol{\delta}_{\text{adv}}$ compared to $\boldsymbol{\delta}_{\text{ran}}$, Table 1(left) tells that the adversarial perturbation can guarantee much stronger denoising effect than random yet they are more expensive for training. Nevertheless, we still note that the impact of our Eq. (10) is remarkable even in the case of random perturbation keeping the FID far below the baselines.

**Resistant to noise by design.** Table 2 compares our approach with the baseline DDPM and DDIM on CIFAR-10 yet corrupted with Gaussian noise. We start by showing that, despite our method has an inductive bias towards denoising the data distribution, if we apply it to the original dataset with no noise ($p = 0\%$), we only get a slight increase in the FID—from 7.2 to 28.66. Yet, if we inspect the results visually we discover that ours is actually smoothing the background of CIFAR-10 and object shape and outlines are still visible as shown in Fig. 4. More results are available in the supplementary material. When we switch to noisy settings, we have

DDPM (Ho et al., 2020)             **Robust**adv

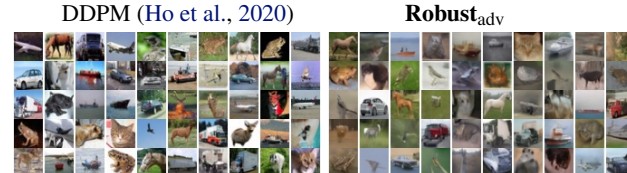

*Figure 4.* Despite the FID decreases once trained on clean data, generated images by **Robust**adv look smooth and the clutter in the background has been canceled.

a large improvement over the baseline for both DDPM and DDIM. We highlight that while the baseline FIDs skyrocket to very high values for $p = 90\%$, $\sigma = 0.2$, the robust DMs are able to keep it in a reasonable range: the DDIM FID de-

*Table 1.* (*left*) Random vs Adversarial Perturbations (*right*) Robust diffusion flow allows less sampling steps keeping high fidelity.

| $\sigma$ | 0.1 | | 0.2 | | Steps | 300 | | 500 | |
|---|---|---|---|---|---|---|---|---|---|
| metrics | FID | IS | FID | IS | metrics | FID | IS | FID | IS |
| **Robust**$_{ran}$ | 79.21 | 5.21 | 68.04 | 4.34 | DDPM | 224.38 | 3.33 | 28.07 | **8.46** |
| **Robust**$_{adv}$ | 24.70 | 7.21 | 24.81 | 7.07 | **Robust**$_{adv}$ | 37.89 | **6.39** | 24.34 | 7.53 |

*Table 2.* Evaluation of performance under different noise conditions for CIFAR-10 using FID ↓ and IS ↑ for DDPM and DDIM.

| p% → | 0 | | 0.9 | | | |
|---|---|---|---|---|---|---|
| $\sigma$ → | 0 | | 0.1 | | 0.2 | |
| metrics → | FID | IS | FID | IS | FID | IS |
| DDPM (Ho et al., 2020) | **7.2** | **8.95** | 58.05 | 6.93 | 102.68 | 4.19 |
| **Robust**$_{adv}$ | | 28.68 | 7.04 | **24.70** | **7.21** | **24.81** | **7.07** |
| DDIM (Song et al., 2020) | **11.62** | **8.36** | 59.28 | **6.89** | 105.43 | 4.09 |
| **Robust**$_{adv}$ | | 31.20 | 6.38 | **25.48** | 6.85 | **24.93** | **6.69** |

parts to 105.43, whereas ours manages to keep it below 25. We offer a qualitative comparison in Fig. 8 where we can notice the benefits of our method. More images available in the supplementary material.

**Time complexity.** Training with AT as an impact during the training time, in particular, we estimate a slow down of ×2.5 for **Robust**$_{adv}$ whereas **Robust**$_{ran}$ is less time-consuming since it does not have to backprop for the adversarial perturbation. Despite the training time being lower than the baseline, remarkably the inference time is the same as other methods and we can attain faster sampling—see Section 3.3.

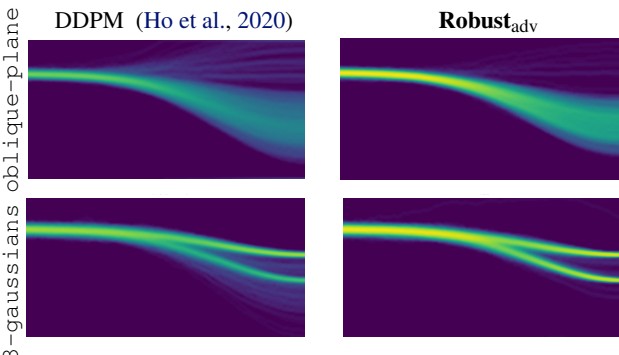

*Figure 5.* Diffusion flow of DMs vs Robust DMs. Left column shows the result by Ho et al. (2020) affected by outliers. Regular training tends to incorporate the noise inside the diffusion flow, making it more prone to generate undesirable and unexpected results; Right column is **Robust**$_{adv}$ that trades off variability for resilience with heatmaps more concentrated, clear, and less faded.

### 3.2. Robust DMs memorize less

Following Daras et al. (2024d) we show that Robust DMs are naturally less prone to memorize the training data. We perform an experiment following Somepalli et al. (2023):

using DDPM and our **Robust**$_{adv}$ trained on clean CIFAR-10, we synthesize 50K images from each of them and measure the similarities of those images with the one in the training set, embedding the images with DINO-v2 (Oquab et al., 2023). In Daras et al. (2024d) a similar experiment was done yet using DeepFloyd IF instead of U-Net DDPM. Although U-Net has much less parameters than DeepFloyd IF—millions vs billions—one could assume that U-Net will overfit less. Instead from Fig. 6 it can be observed that there is still a decent amount of generated samples with similarity higher than 0.90. A similarity ≥ 0.9 roughly corresponds to the same CIFAR image. Robust models have a histogram that is drastically shifted on the left and the curve of the histogram in the right part decays more rapidly than DDPM. In the region ≥ 0.9, ours have far less training replicas.

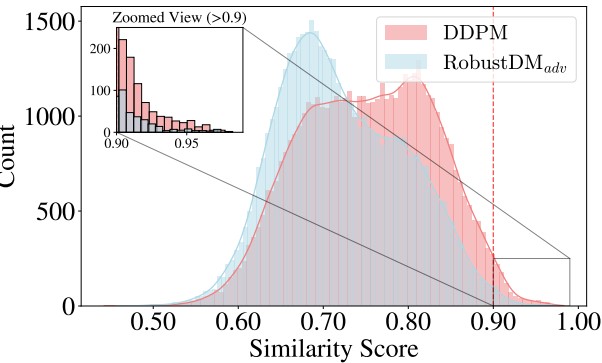

*Figure 6.* Histogram of similarities between generated samples from DMs and the CIFAR-10 training set. Similarity values above 0.9 roughly correspond to the same image with some variation in color, orientation, or background details. While values above 0.94 suggest nearly identical images. DDPM generated images (red) are closer to the training set. In contrast, **Robust**$_{adv}$ (blue) shows a noticeable shift to the left and drastically reduces the replicas.

### 3.3. Smooth Diffusion Flow enables Faster Sampling

**Smooth Diffusion Flow.** This section describes the results obtained by applying our method to the classical DDPM (Ho et al., 2020). Fig. 5 shows the diffusion flow mapping the standard normal distribution to the data distribution from left to right. To visualize the flow, we use low-dimensional 3D data and every time project the data in 2D by simply removing a coordinate. In `oblique-plane`, we can see how Robust DMs capture less variability in the dataset, thereby rejecting a lot of outlier noise while the heatmap of DDPM is more faded. Moreover, DDPM is misled by the noise and creates a very sublet yet additional mode that is not present in the data, whereas ours keeps the generation unimodal. The same remarks hold for a multi-modal dataset: in `3-gaussians`, DDPM trajectories are misled by noise and fade while ours are kept straight. In this case, the figure shows two modes: one is not displayed due to projection.

**Faster Sampling.** Fig. 5 shows that the diffusion flow of a Robust DM is more compact, less faded and more dense. This could imply that the inference process may still recover the right path in case the regressed score vector is corrupted or is noisy or in case we deliberately use fewer steps in Eq. (5) for faster sampling. We tested this hypothesis and the trade-off table of FID in function of the number of steps taken is shown in Table 1(right). Even more, if we cross compare Table 1(right) with Table 2, on clean data **Robust**$_{adv}$ scores a better FID with 500 steps (24.34) vs 1000 steps (28.68). This experiment supports our claim showing that Robust DMs are still able to generate samples with good fidelity even though we use fewer steps in the inference. The degradation using less steps is widely more graceful than DDPM especially when we take only 300 steps over 1000.

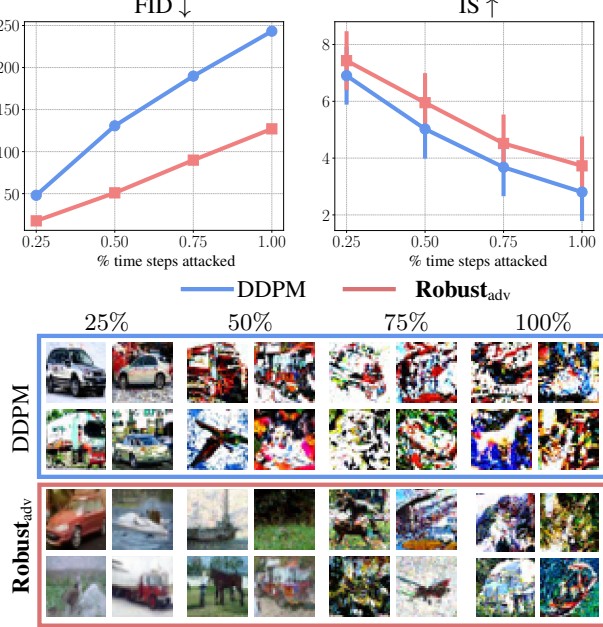

*Figure 7.* Robustness to Adversarial Attacks. While the baseline model DDPM is susceptible to adversarial attacks, Robust DMs are able to better resist to them yielding superior FID and IS for different percentage of time steps attacked: i.e. 25% means that we attack 250 steps uniformly over the 1000 of DDPM.

### 3.4. Robustness to Adversarial Attacks

Our method is naturally resilient to attacks. Like classifiers, AT enforces robustness to adversarial perturbations in the diffusion flow. The attack has to take into account the stochastic nature of DM inference and the fundamental hypothesis of Gaussianity for each diffusion stage. We attack a DM in a white-box setting by adding, in *some* of the intermediate steps of the inference, an adversarial perturbation defined as described in Algorithm 2. We need to be careful with the range of values of the perturbation to maintain the assumption of the diffusion process; more information can

---

**Algorithm 2** Adversarial Attack on a Diffusion Model.

**Input:** percentage of attacked timesteps $p$, total timesteps $T$, model $\epsilon_\theta$, scheduler values $\alpha_t$ and $\sigma(t)$, perturbation strength $\phi$; $\mathbf{x}_T \sim \mathcal{N}(0, I)$
**for** t $= T$ to 0 **do**

$\quad \mathbf{x}_{t-1} \leftarrow \epsilon_\theta(\mathbf{x}_t, t)$

$\quad \hat{\mathbf{x}}_0 \leftarrow \frac{\mathbf{x}_t - \sqrt{1-\bar{\alpha}_t}\epsilon_\theta(\mathbf{x}_t,t)}{\sqrt{\bar{\alpha}_t}}$

$\quad \tilde{\boldsymbol{\mu}}_t(\mathbf{x}_t, \hat{\mathbf{x}}_0) \leftarrow \frac{\sqrt{\bar{\alpha}_{t-1}}\sigma(t)}{1-\bar{\alpha}_t}\hat{\mathbf{x}}_0 + \frac{\sqrt{\alpha_t}(1-\bar{\alpha}_{t-1})}{1-\bar{\alpha}_t}\mathbf{x}_t$

$\quad \mathbf{x}'_t = \mathbf{x}_t + \boldsymbol{\delta}; \quad \boldsymbol{\delta} \sim \mathcal{N}(0, I) \cdot \phi \cdot \sigma(t)$

$\quad \mathbf{x}'_{t-1} \leftarrow \epsilon_\theta(\mathbf{x}'_t, t)$

$\quad \hat{\mathbf{x}}'_0 \leftarrow \frac{\mathbf{x}'_t - \sqrt{1-\bar{\alpha}_t}\epsilon_\theta(\mathbf{x}'_t,t)}{\sqrt{\bar{\alpha}_t}}$

$\quad L = \left\| \tilde{\boldsymbol{\mu}}_t(\mathbf{x}_t, \hat{\mathbf{x}}_0) - \tilde{\boldsymbol{\mu}}_t(\mathbf{x}'_t, \hat{\mathbf{x}}'_0) \right\|_2^2$

$\quad \mathbf{x}_t^{adv} = \mathbf{x}_t + \boldsymbol{\delta} \quad \text{where} \quad \boldsymbol{\delta} = \sigma(t) \cdot \text{sign}(\nabla_{\mathbf{x}_t}\mathcal{L}_{DM})$

$\quad \mathbf{x}_{t-1} \leftarrow \epsilon_\theta(\mathbf{x}_t^{adv}, t)$

**end for**

---

be found in Appendix A.1. Fig. 7 shows that our method is much more robust to attacks in diffusion flow: robust DMs can tolerate up to 50% of time step attacked and still generate samples with decent fidelity. Only at 75% time steps attacked, the generation fails for both.

## 4. Related Work

**Diffusion Models.** Score-based generative models (Song & Ermon, 2019) introduce a maximum likelihood framework focused on learning the score function–the gradient of the log density–to generate samples that are likely to be part of the real data distribution. These models express the inference process through a Stochastic Differential Equations (SDE) (Dhariwal & Nichol, 2021). Non-equilibrium thermodynamics inspired the diffusion process (Sohl-Dickstein et al., 2015), a concept that from particle physics was applied as a peculiar case of score-based generative model and that led to the rise of Denoising Diffusion Probabilistic Models (DDPMs) (Ho et al., 2020). Diffusion models (DM) became the *de-facto* standard algorithm in generative modeling on high-dimensional data, overcoming the previous adversarial min-max game between a generator and a discriminator (GANs) (Goodfellow et al., 2020). DMs not only achieve higher fidelity (Dhariwal & Nichol, 2021) but also provide more stability in training. DMs have been extensively improved: working on the logarithmic likelihood estimate (Nichol & Dhariwal, 2021), faster sampling (Nichol & Dhariwal, 2021; Song et al., 2021a) and performing the diffusion process in the latent space (Rombach et al., 2022) instead of the data space. In Karras et al. (2022; 2024), the authors provide insightful clarifications on several design choices for diffusion models. Furthermore, they introduce an improved U-Net architecture featuring redesigned network layers that ensure consistent activation, weight, and

DDPM (Ho et al., 2020)                    **Robust**adv

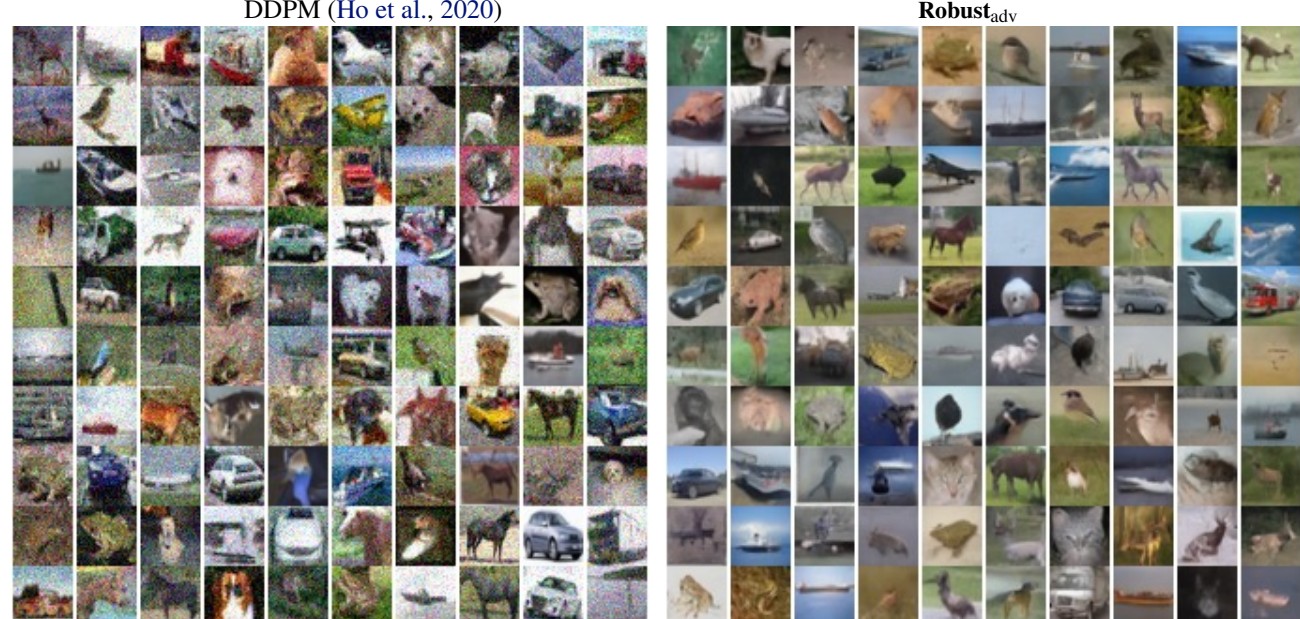

*Figure 8.* Qualitative comparison. We offer qualitative samples of **Robust**adv vs DDPM. Despite the training containing 90% of corrupted data with strong Gaussian noise with $\sigma = 0.2$, **Robust**adv generates smooth objects with no visible noise or artifacts while DDPM outputs the Gaussian noise. $\sigma = 0.2$ means we are adding 40% of the variability that is naturally present in CIFAR-10 being $\sigma_{\text{data}} = 0.5$.

update magnitude, achieving state-of-the-art FID on CIFAR and other benchmarks. Lastly, Song et al. (2023) proposed *consistency* models, a distillation method for one-step inference by directly mapping noise to data. The name *consistency* arises from the fact that they enforce different noisy versions in the same trajectory to map to the same data. Unlike them, we enforce *smoothness* in the local neighborhood of a trajectory so its score field remains locally consistent.

**Denoising and Inverse problems with DMs.** The attention to applying DMs in scenarios where the data is not assumed to be curated and clean but is instead degraded or corrupted, has increased in recent years (Aali et al., 2023; Xiang et al., 2023; Daras et al., 2024d;a). Given the specific challenges related to training with noisy data, this problem is closely related to inverse problems (Tachella et al., 2024; Kawar et al., 2024). Recently, a line of research has focused on the application of Stein's Unbiased Risk Estimator (SURE) (Metzler et al., 202) and its subsequent improvements, including UN-SURE (Tachella et al., 2024), GSURE (Kawar et al., 2024), Soft Diffusion (Daras et al., 2024b), and methods leveraging optimal transport for training with noise (Dao et al., 2024).

**Adversarial Robustness.** Adversarial robustness is loosely connected with denoising since adversarial training (AT) can be viewed as a means of removing spurious correlations (Ye et al., 2024) with improved out-of-domain generalization when transferring to a new domain (Ilyas et al., 2019) or related to causal learning (Zhang et al., 2020; 2022). Surprisingly, only a few papers show that AT actually increases spu-

rious correlations in some cases (Moayeri et al., 2022a;b), although robustness tools are still used to assess if a neural model relies on spurious associations between the input and the output class (Singla & Feizi, 2022; Neuhaus et al., 2023). AT variants have also been used to improve domain shift (Salman et al., 2020) and out of distribution (Wang et al., 2022). While it is reasonable to say that AT has been extensively studied on discriminative classifiers and GANs, its application to DMs remains relatively unexplored, except for Sauer et al. (2024) in which it is applied for fast sampling and Yang et al. (2024) which investigates the interconnection between samples in a batch.

## 5. Conclusions and Future Work

We have presented the first attempt to incorporate AT into DM training showing that AT for generative modeling implies smoothing the data distribution and can be effectively used for denoising the data. We also have shown that we need to reinterpret it as *equivariant* property and not *invariance*. Our method has been proved to be highly robust even if presence of 90% corrupted data with strong Gaussian noise. For future work we have plenty of ideas: in the encoding steps of the denoising we believe we can improve the encoding functions by letting the network learn an $\omega$ that is input dependent. We also have to extend our method to work in fully corrupted settings ($p = 100\%$) and port our approach to Elucidating DM framework (EDM) (Karras et al., 2022; 2024) in order to scale to larger datasets.

## Impact Statement

*This paper presents work whose goal is to advance the field of Machine Learning, in particular the robustness of a particular type of generative models, Diffusion Models, that are widely used in the industry and for content creation. There are many potential societal consequences of our work: we believe the robustness of DMs is not well studied and can have highly beneficial societal impact. For example, we can consider as outliers NSFW content or other minor modes that are not proper of latent factors that we wish to learn. Our research can enable generative AI which is more robust as less susceptible to adversarial perturbations and memorizing less the data having models that better respect privacy.* **This field is mandatory and does not count in the page limit**

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

# A. Appendix

## A.1. Attack formulation

In inference mode, it is possible to represent the inverse Markov Chain as the sequence of intermediate realizations of Gaussian distributions with fixed parameters regarding mean scaling and variance scaling. From the paper (Ho et al., 2020) in Eqs. 6 and 7 the $t$-th step of the inference can be written as the sampling from the posterior distribution $q(\mathbf{x}_{t-1}|\mathbf{x}_t t, \mathbf{x}_0) = \mathcal{N}(\mathbf{x}_{t-1}; \tilde{\boldsymbol{\mu}}_t(\mathbf{x}_t, \mathbf{x}_0), \tilde{\beta}_t \boldsymbol{I})$, where:

$$\tilde{\boldsymbol{\mu}}_t(\mathbf{x}_t, \mathbf{x}_0) := \frac{\sqrt{\alpha_{t-1}}\sigma(t)}{1 - \alpha_t}\mathbf{x}_0 + \frac{\sqrt{\alpha_t}(1 - \alpha_{t-1})}{1 - \alpha_t}\mathbf{x}_t, \quad \tilde{\beta}_t := \frac{1 - \alpha_{t-1}}{1 - \alpha_t}\sigma(t).$$

This implies that, at each time step, the expected variance and mean of the distribution are defined in a specific manner. During inference, the value of $\mathbf{x}_0$ corresponds to the output obtained after the network's prediction. In the context of the DDPM framework, $\mathbf{x}_0$ is replaced by the estimated value, which depends on the epsilon-predicting network:

$$\hat{\mathbf{x}}_0 = \frac{\mathbf{x}_t - \sqrt{1 - \alpha_t}\boldsymbol{\epsilon}_\theta(\mathbf{x}_t)}{\sqrt{\alpha_t}},$$

To properly craft the attack, and still consider it legitimate, it is essential to scale it to the correct standard deviation to align with the diffusion process. Failing to do so would result in the network's inference being affected not by the perturbation itself but by the incorrect range of the perturbation, causing errors due to the inability to maintain the process within its Gaussian assumptions.

In this context, the attack procedure follows the FGSM approach with random start. However, the perturbation is then scaled to match the appropriate variance at timestep $t$ to maintain consistency with the diffusion process.

The FGSM attack generates an adversarial example by perturbing the noisy sample $\mathbf{x}_t$ in the direction of the gradient of the loss function $\mathcal{L}$ with respect to $\mathbf{x}_t$. Specifically, the adversarial perturbation is given by:

$$\mathbf{x}'_t = \mathbf{x}_t + \phi \cdot \text{sign}\big(\nabla_{\mathbf{x}_t}\mathcal{L}_{\text{DM}}(\mathbf{x}_t)\big),$$

where $\phi$ controls the magnitude of the perturbation, $\text{sign}(\cdot)$ represents the element-wise sign function.

The adversarial attack in this approach is integrated into the diffusion process by leveraging the predictive functions including a variance-handling mechanisms defined in the model in order to guarantee to concretely adapt to the Gaussian hypothesis of the reverse MC. The adversarial attack begins with perturbing the input $\mathbf{x}_t$ defining its $\mathbf{x}'_t$ as:

$$\mathbf{x}'_t = \mathbf{x}_t + \boldsymbol{\delta}, \qquad \boldsymbol{\delta} \triangleq \mathcal{N}(0, \phi \cdot \sigma(t)).$$

The function to be optimized in order to craft the adversarial attack is defined as follows:

$$\mathcal{L}_{FGSM} = \big\|\tilde{\boldsymbol{\mu}}_t(\mathbf{x}_t, \mathbf{x}_0) - \tilde{\boldsymbol{\mu}}_t(\mathbf{x}'_t, \mathbf{x}_0)\big\|_2^2$$

where $\tilde{\boldsymbol{\mu}}_t$ represents the predicted mean of the diffusion process at time step $t$, which depends on both the input $\mathbf{x}_t$ and the original sample $\mathbf{x}_0$. The optimization goal is to maximize the discrepancy between the predicted means of the clean and adversarial inputs, ensuring that the perturbation effectively disrupts the reverse diffusion process.

To compute the adversarial perturbation $\boldsymbol{\delta}$, the gradient of the loss $\mathcal{L}_{FGSM}$ with respect to $\mathbf{x}'_t$ is used:

$$\boldsymbol{\delta} = \sigma(t) \cdot \text{sign}\left(\nabla_{\mathbf{x}_t}\mathcal{L}_{FGSM}\right),$$

where $\sigma(t)$ scales the perturbation to ensure it adheres to the variance of the Gaussian noise in the reverse diffusion process at the $t$-th step. This step aligns the adversarial attack with the stochastic nature of the model, ensuring the perturbation remains consistent with the Gaussian hypothesis.

The final adversarial example is then obtained as:

$$\mathbf{x}_t^{\text{adv}} = \mathbf{x}_t + \boldsymbol{\delta}.$$

The adversarially perturbed sample $\mathbf{x}_t^{\text{adv}}$ is fed back into the reverse diffusion process, following the recurrency of the inference.

