# OpenReview forum: "What is Adversarial Training for Diffusion Models?"
_ICML.cc/2025/Conference — Submitted to ICML 2025_

### Official Review · Reviewer_a6SF · 2025-03-12

**Overall Recommendation:** 3

**Summary:**

This paper investigates AT tailored specifically for DMs, emphasizing that adversarial robustness for DMs should enforce equivariance rather than invariance. The authors introduce a new approach where perturbations, either random or adversarial, are added to enforce smoothness in the diffusion trajectories. Empirical evaluations on synthetic and real datasets (such as CIFAR-10) demonstrate that their method effectively enhances robustness, reduces memorization, and improves the resilience of diffusion models under various types of noise and adversarial attacks.

**Claims And Evidence:**

The current experimental evaluation appears limited, primarily relying on synthetic and relatively small-scale datasets, thus leaving questions regarding the generalizability and scalability of the proposed approach largely unanswered. To strengthen the claims, it would be beneficial to evaluate the method on larger-scale datasets or real-world noisy datasets, thereby providing stronger and more convincing evidence of practical robustness and broader applicability.

**Essential References Not Discussed:**

The authors have discussed relevant references clearly, but as mentioned earlier, it would be beneficial to also include references to general adversarial defense techniques that utilize denoising strategies, even if they are not specifically designed for diffusion models. For example, works focusing on adversarial purification or denoising-based robustness methods (such as those based on randomized smoothing or explicit denoising modules) could be relevant for broader context and are currently not adequately discussed.

**Experimental Designs Or Analyses:**

The experimental setup and analyses appear generally sound, appropriate, and sufficiently clear.

**Methods And Evaluation Criteria:**

The authors’ approach of defining AT for diffusion models through equivariance is conceptually sound and makes sense given the generative nature of these models.

**Other Comments Or Suggestions:**

N/A

**Other Strengths And Weaknesses:**

Overall, the paper is clearly structured and easy to follow. The conceptual logic is presented clearly, and the theoretical analysis provided is insightful and thorough. The main limitation lies in the experimental evaluation, which remains somewhat narrow in scope and dataset complexity. Moreover, comparisons or discussions with other adversarial robustness methods that utilize denoising or smoothing strategies are currently insufficient, and addressing this gap would significantly strengthen the manuscript.

**Questions For Authors:**

Can you elaborate on how the proposed method compares to or differs from existing adversarial defense methods that also employ denoising or randomized smoothing strategies? Including a discussion or comparison could clarify the uniqueness and significance of your approach.

**Relation To Broader Scientific Literature:**

The authors have clearly discussed relevant work specific to diffusion models; however, it would be beneficial if they could additionally consider related adversarial robustness techniques from a broader perspective. Specifically, several adversarial defense methods have previously leveraged denoising strategies to improve robustness, even though not limited to diffusion models. Including or comparing against these methods might further strengthen the paper by placing it in a wider context of adversarial robustness literature.

**Theoretical Claims:**

The formulations appear generally sound, and I did not notice significant issues.

---

> ### Author Rebuttal · Authors · 2025-03-31
>
> We thank the reviewer for the feedback and constructive criticism. We are glad the reviewer mentioned that our method and analysis are sound, appropriate, and sufficiently clear, giving comments such as: the paper is clearly structured and easy to follow; that the conceptual logic is presented clearly, and the theoretical analysis provided is insightful and thorough.
>
> We hope to answer the final remarks:
>
> **1. Generalization to large scale dataset - rev `a6SF`; Effectiveness on more complex datasets - rev `xC6d`**
>
> We show our method is still effective on more complex datasets, covering a wide range of cases: more samples, more classes, higher resolution. We will add them to the revised version. After corrupting the data as described in L.262-267-left, we provide results on:
>
> - **CelebA** that consists of 202K face images of 64x64 pixels (x4 CIFAR-10 cardinality, x2 resolution), being it a benchmark dataset for image generation task;
> - **TinyImageNet** 100K samples at 64x64 resolution with 200 classes (x20 more classes than CIFAR-10). TinyImageNet is not a benchmark for generation, we tested the faster DDIM only at $\sigma=0.2$ due to the time limit.
>
> | Method $\downarrow$ |  CelebA | CelebA
> |--|--|--|
> | Noise $\rightarrow$ |  $\sigma=0.1$ | $\sigma=0.2$
> | DDPM 		| 56.72 		| 96.08
> | Robust$_{\text{adv}}$ - DDPM 	| **14.4**  	| **16.4**
>
>
> | Method $\downarrow$ |  TinyImageNet
> |--|--|
> | Noise $\rightarrow$ |  $\sigma=0.2$ |
> | DDIM 		| 66.2 		|
> | Robust$_{\text{adv}}$ - DDIM 	| **48.9**  |
>
> We provide qualitative results of these experiments, showing our generation vs the baselines:
>
> - [Qualitative comparison on CelebA](https://bashify.io/i/CPm0D6)
> - [Qualitative comparison on TinyImageNet](https://bashify.io/i/bpcTIN )
>
> **The same outcome we have in the paper transfers to these two datasets**: For both of these new datasets, we observe the same pattern as before. Our approach induces smoothness in the data distribution thereby removing the noise in case it is present. Despite the Gaussian noise increasing, our method still keeps the FID lower than the baseline. This is achieved by trading robustness for variability: it still generates images that are natural yet more “smooth” with less cluttered background or it makes abnormal samples more “normal”. For instance, in CelebA, Robust DMs avoid generating faces that resemble outliers: faces where image quality is so low that the data does not look like a canonical face anymore. The same applies to TinyImageNet yet the baseline is weaker since we used the same network as in CIFAR with a larger input size and no hyperparameter tuning due to time constraints. Still the model trades off learning variability for improved smoothness. Indeed in the qualitative images above we see that, despite Robust DMs generating smoother faces, they also remove the added Gaussian noise.
>
> **2. Discussions of existing adversarial defenses with denoising or randomized smoothing**
>
> In response to rev. `riys` and rev. `a6SF`, we will add a mini-section at the end of Section 4 discussing adversarial defense techniques, including randomized smoothing–e.g., Cohen et al. (2019)–and denoised smoothing [C,D]. For rev. `a6SF`, please see **point 1** in the response to rev. `riys`.
>
> **We also clarify that in our case denoising is not related to randomized smoothing but means handling noise in the dataset when training DMs, thereby denoising the data distribution.**
>
> While many of such methods were originally designed for classifiers, ours is specifically designed to make the DM robust in the generation. Following rev. `riys`, we now highlight adversarial purification (AP) methods that fine-tune generative models used for purification for improved robustness and cite [1], as their objectives align with our work despite differences in how adversaries are generated [1,E,F]. In the context of AP, recent work [G] shows that adversarial perturbations disrupt the generative process of diffusion models, causing deviations from the clean trajectory. While our current focus does not extend to adversarial purification, we note that enforcing local smoothness in the score field might help mitigate such deviations. In fact, interpreting our work along with a classifier is very interesting because this could lead to future work applying our Robust DMs to AP. Due to space constraints, here we list only a few key references but will be happy to add any you suggest in the next discussion phase.
>
> [C] (Certified!!) Adversarial Robustness for Free! ICLR 2023
>
> [D] Denoising Masked Autoencoders Help Robust Classification ICLR 2023
>
> [E] Towards Understanding the Robustness of Diffusion-Based Purification: A Stochastic Perspective ICLR 2025
>
> [F] ADBM: Adversarial Diffusion Bridge Model for Reliable Adversarial Purification ICLR 2025
>
> [G] MimicDiffusion: Purifying Adversarial Perturbations via Mimicking Clean Diffusion Models CVPR 2024

---

### Official Review · Reviewer_xC6d · 2025-03-13

**Overall Recommendation:** 3

**Summary:**

This work endeavors to construct a novel adversarial training approach for diffusion models. Through comparing the AT process of traditional classification models, the author suggests that the crucial key of DMs AT resides in equivariance. Consequently, the perturbation process and adversarial training loss in the reverse process are derived. After verification on CIFAR-10 data, the effectiveness of the model is demonstrated.

**Claims And Evidence:**

The authors have essentially provided sufficient evidence to support their motivation. However, there is one aspect that I am not entirely clear about: How exactly does Figure 2(a) reflect robust information? What information do the colored line clusters signify? Particularly in the case of Strong Inliers in the upper portion, I did not detect any obvious distinction between DDPM and Robust adv.

**Essential References Not Discussed:**

The paper's references in the work are sufficient.

**Experimental Designs Or Analyses:**

1. This work has not been validated on more complex datasets like Imagenet or TinyImagenet. Is it possible to attain effective robustness on these datasets by fine-tuning or AT on such datasets?
2. Table 2 shows that the performance of the author's method is substantially lower than that of the baseline, DDPM, and DDIM or even lower than Adversarial Perturbation performance. In traditional AT, although it may lead to a certain decline in clean performance, the clean performance is typically still higher than that of adv. Does this work lack generalization to clean samples?
3. The author focuses on the evaluation process in Chapter 3 of the paper. In Chapter 3 of the Appendix, does the author's training process involve direct initialization training or fine-tuning on an existing diffusion model? How does the cost of AT compare to ordinary training? Can it be compared in terms of time or computation?

**Methods And Evaluation Criteria:**

1. In Formula 7, δ is utilized in Algorithm 1. However, δ_{adv} in Formula 8 does not make an appearance in the algorithm.  So how δ_{adv} functions.
2. In "Key change is equivariance," the author proposed a new AT loss expression as ||eps_theta(x_adv)-(eps+delta)||. Additionally, the loss expression at the end of formula 10 becomes ||eps_theta(x_adv)-(eps_theta+delta)||. Is there any inconsistency between eps and eps_theta?
3. I did not observe any references to the attack method employed in Algorithm 2. Is this an attack algorithm constructed in this work? Does there exist a public and normally used method for conducting adversarial attacks on diffusion? Can the authors conduct verification on it?

**Other Comments Or Suggestions:**

There is no other comment.

**Other Strengths And Weaknesses:**

The author's presentation of the experimental results and insights is highly professional and elegant. Although a sufficient number of subjective and objective experiments have been provided on CIFAR-10, there is still a relative dearth in the selection of natural image datasets (with experiments only being conducted on CIFAR-10).

**Questions For Authors:**

1. Demonstrate the effectiveness of the method on more complex datasets.
2. Clarify the description of the method.

**Relation To Broader Scientific Literature:**

This work provides a robust diffusion model training method, which may be helpful for improving the robustness of the widely used stable diffusion.

**Theoretical Claims:**

Proofs for theoretical claims are complete.

---

> ### Author Rebuttal · Authors · 2025-03-31
>
> Thank you for the feedback and constructive criticism. We are glad the reviewer acknowledged that we provided sufficient evidence to support our motivation, our method could be helpful for the future robustness of SD and our experimental results and insights are highly professional and elegant. Below we respond to the remaining remarks:
>
> **1. Per reviewer request, below we provide clarification to the description of the method.**
>
> > How exactly does Figure 2(a) reflect robust information?
>
> As mentioned at L.74-77 of our contribution, we are the first to reconnect AT for DMs as a denoising method, so Fig.2(a) shows how DMs, trained with AT, behave when their training is affected by data noise, e.g., points with either strong inlier noise or uniform outliers. DDPM is attracted by noisy points and generates both the plane *and* the noise whereas ours generate *only* the plane *despite being trained on the same noisy points*.
>
> > What information do the colored line clusters signify?
>
> The colored lines are the 3D trajectories of the generated points w/ Eq.(5). Starting from $\mathbf{x}\_0 \sim \mathcal{N}(0,1)$, the trajectories flow toward the learned $p_{\theta}(\mathbf{x})$. Each color is a different sampling. We will add this explanation to L.132 second column when citing Fig.2(a).
>
> > *Fig. 2(a)-”Strong Inliers”*, no distinction DDPM vs Robust
>
> Zooming in, DDPM generates trajectories going behind or out of the plane, i.e. cyan-colored trajectory in the *opposite direction of the plane*. Ours does not produce those diverging trajectories, better reaching  the center of the plane.
>
> **Trading variability for robustness**: while we obtain robustness, we capture less variability that is in reality present in the data distribution. In other words we trade off variability for smoothness. A similar logic is mentioned in the caption of Fig.2(b) at L.124.
>
> > In Formula 7, δ is utilized in Algorithm 1. However, δ_{adv} in Formula 8 does not make an appearance in the algorithm. So how δ_{adv} function?
>
> Thank you. This is a typo on our side that we will fix. L.227 of Algo.1 should be using $\boldsymbol{\delta}\_{\text{adv}}$, optimized as Eq.(8). We also need to update Eq.(6) with delta with adv. In a nutshell, delta with adv. represents the perturbation *after* the gradient step, at the end of optimization while delta is the random initialization.
>
> > Is there any inconsistency between eps and eps_theta?
>
> At L.211-right there is a single loss that regresses both \eps and \delta, doing so we enforce equivariance yet not smoothness in the prediction, for smoothness you need \eps to be \eps_theta so that is a function of both weights and input x_t. In this sense, the regularization–Eq.10–can be seen as two outputs of the same network interacting together to enforce smoothness (L.233-234). We will clarify this in the revision.
>
> > Is Algo. 2 constructed in this work? There exist normally used attacks on DMs?
>
> The attack is constructed in this work as an analytical tool for the worst-case robustness of DMs. We verified and in contrast to existing attacks that compute a single input perturbation and thus perturb the DM trajectory in an “averaged” sense, ours computes an optimal perturbation at every time step that maximizes the DM loss. It demonstrates that there exists a sequence of perturbations that could maximally disrupt the trajectory at each inference step. Such an analytical framework is valuable for understanding the fundamental sensitivity of the generative process.
> Existing attacks have been predominantly developed for latent DMs, and do not transfer well to pixel-based DMs [B].
> >  Although traditional AT may lead to a certain decline in clean performance, the clean performance is typically still higher than that of adv. Does this work lack generalization to clean samples?
>
> The clean performance is better than under attack in our case too. If we attack a good number of time steps, the FID is worse than without attack–see Fig. 7. For generalization without attack, please see the discussion above on **trading variability for robustness**: Tab. 2 is without attacking the DM, in that case we have better FID (~24) when we train with data corrupted at sigma=0.1 because we smooth out the noise whereas in the case of clean data (sigma=0), we smooth some “part” of the data distribution (FID ~28).
>
> **2. Fine-tuning vs full training; Computational cost.**
>
> We always trained from scratch thereby avoiding fine-tuning. A time complexity overhead estimate is given at L.298-305. DDPM operations are a single forward for getting the prediction and a backward pass for weights update. With our regularization, we have to add a backward pass for getting the gradients of the adversarial loss over the perturbation, then repeat the DDPM operations above.
>
> **3. Effectiveness on more complex datasets.** Please see response to rev. `a6SF`.
>
> [B] Pixel is a Barrier: Diffusion Models Are More Adversarially Robust Than We Think, NeurIPSW 2024

---

### Official Review · Reviewer_riys · 2025-03-15

**Overall Recommendation:** 3

**Summary:**

This work studies adversarial training for diffusion models, highlighting its fundamental differences from adversarial training for classifiers. Unlike adversarial training for classifiers enforcing invariance, adversarial training for diffusion models requires equivariance to ensure the diffusion process remains within the data distribution. The proposed method enforces smoothness in the diffusion flow and aims to improve robustness to noise, outliers, and data corruption without relying on specific noise assumptions. Experimental results also demonstrate its effectiveness.

**Claims And Evidence:**

Yes, in general, the claims are supported by clear and convincing evidence. However, there are two claims not being articulated with clarity.

1. *Claim 1*: Classifiers require invariance under adversarial perturbations, but DMs demand equivariance so that the diffusion process remains within the data distribution.

    *Evidence 2*: Section 2.4 provides the analysis and the explanations for evidence. However, they are not strong enough theoretically and empirically. The performance of the proposed method is not compared with the naive adversarial training of diffusion models as stated in Equation (9).

2. *Claim 2*: The proposed adversarial training method makes diffusion models more robust to adversarial attacks.

    *Evidence 2*: Experimental design in Section 4 to evaluate this robustness is not comprehensive enough. There is only on white-box attack: FGSM applied, where the other common but complicated adversarial attacks are ignored, such as PGD and AutoAttack. Hence *Claim 2* is not convincingly supported.

**Essential References Not Discussed:**

When discussing the adversarial training for diffusion models, especially for enhancing the robustness, one key paper should be discussed on adversarial training for diffusion on purification (AToP) [1] which is also on the similar task, in the introductions. Hence the comprehensiveness of this work can be improved.

[1] G. Lin, C. Li, J. Zhang, T. Tanaka, and Q. Zhao, Adversarial training on purification (AToP): Advancing both robustness and generalization, ICLR2024.

**Experimental Designs Or Analyses:**

To the best of my knowledge, I find the experimental design not very sound. Although it evaluates the performance of the proposed algorithm to apply adversarial training in diffusion models for denoising, why are the adversarial attacks are chosen to be FGSM, as stated in Appendices? If the adversarial training is via PGD, does it change the performance?

Also, I am wondering if diffusion models are successfully attacked, where the generated images are all wrongly classified, will the FID and IS be influenced?

**Methods And Evaluation Criteria:**

As stated in **Claims And Evidence*, the proposed adversarial training seems to partially make sense for diffusion models, in terms of generative power improvement. It is natural and well articulated with changed similarity that adversarial training can handle the memorization issue in diffusion models.

However, I am still not convinced that the robustness can be evaluated by FID and IS when defending against adversarial attacks. Please see the second question in **Experimental Designs Or Analyses**.

**Other Comments Or Suggestions:**

NA

**Other Strengths And Weaknesses:**

**Strengths**:

1. This work proposed a novel adversarial training method for diffusion models to enhance the robustness and generation quality via adversarial perturbations induced in the forward and reverse processes. It is an original idea which is not sufficiently studied before.

2. The significance of this work is also a highlight. It bridges the generative power and robustness in the training process of diffusion models. Hence it is possible to pave the way for future studies in both areas.

**Weaknesses**:

1. The logic of this work definitely requires to be polished. Hence the clarity will be enhanced in terms of the coherence. As aforementioned, the major claims should be supported by various experimental results or theoretical guarantees.

**Questions For Authors:**

Please see **Experimental Designs Or Analyses**.

**Relation To Broader Scientific Literature:**

This paper provides contributions to robustness, and the generative power diffusion model. It challenges existing understanding and methods for adversarial training via an equivariance-based approach. It also demonstrates empirical improvements in robustness and memorization reduction. However, theoretical justifications and evaluations on stronger attacks would strengthen its impact in the broader scientific context.

**Theoretical Claims:**

There is no theoretical claim in this work.

---

> ### Author Rebuttal · Authors · 2025-03-31
>
> Thank you for the feedback and constructive criticism. We are glad the reviewer appreciated the novelty and original idea that could pave the way for future studies and the significance of our work. Below we respond to the remaining remarks:
>
> **1. Clarification for rev. `riys` and rev. `a6SF`: Our attack on DMs disrupts generation, not a classifier.** We remark that the attacks in our paper are in the trajectory space of DM to disrupt their generative capabilities, not to fool a classifier. See Sec 3.4. Thus, we have chosen SOTA metrics appropriate for DMs (FID/IS).
>
>
> **2. Invariance.**
>
> It was our first failed attempt. We report comparisons across three datasets showing that invariance–replacing the regularization part in Eq.(10) with invariance Eq.(9)–makes the model diverge from the real data distribution $p_{\text{data}}(\mathbf{x})$, actually learning the noise. Results are shown below.
>
> We provide [qualitative results showing the diffusion trajectories of Robust DM trained **with invariance** vs DDPM](https://bashify.io/i/cZymJF), showing invariance actually performs worse than DDPM. Our method is shown in Fig. 2(a) and Fig. 5.
>
> We have [updated Fig. 3(b) with invariance, also trying different $\lambda$ in Eq. (10)](https://bashify.io/i/8a6Rl5) and we will add it to the improved version of the paper.
>
> Finally, we confirm that the FID on CIFAR-10 with invariance is 356.9,much worse even than DDPM.
>
> Fig. 1 in supp. material shows that invariance leads the model to learn a different vector than the desired one. Moreover in Sec. 1.3 of supp. material, the ELBO derivation provides the theoretical explanation supporting the need for equivariance in the training.
>
>
> **3. Robustness to other attacks.**
>
> We ran an iterative attack on DM trajectories, i.e. PGD with 20 iterations. The table **FID under FGSM** below reports Fig.7 in the paper whereas **FID under PGD** reports that with PGD 20 iterations. We see that despite iterating the attack with PGD, ours still maintains a wide gap in the FID compared to the baseline DDPM and PGD does not increase the FID much.
>
>
> **FID $\downarrow$ under FGSM**  (like Fig. 7)
>
> |  %time steps attacked $\rightarrow$ | 25 | 50 | 75 | 100|
> |--|--|--|--|--|
> | DDPM | 49.8 | 131.7  | 190.4  | 243.4  |
> | Robust$_{\text{adv}}$ | **19.29**  | **52.0** | **90.7**  | **127.7** |
>
>
> **FID  $\downarrow$ under PGD** (new)
>
> |  %time steps attacked $\rightarrow$ | 25 | 50 | 75 | 100|
> |--|--|--|--|--|
> | DDPM |  55.7 | 134.5  | 200.3 | 248.1 |
> | Robust$_{\text{adv}}$ | **22.7** | **55.8** | **98.1**  | **128.6** |
>
> Regarding the use of black-box or AA: adapting attacks like SQUARE or FAB-T from classifiers to DMs is very interesting but out of the scope of the paper since they need to be “ported” to DM in a non-trivial way. Nevertheless we will update our future work to mention this line of research.
>
> While some works in adversarial purification (AP) focus on attacking simultaneously DMs and classifiers, these approaches are tailored to classification. In contrast, we study the robustness of generative modeling capabilities of DMs rather than targeting their role in purification. **See point 1.** More information in the reply to rev.`a6SF` at **point 2.**
>
>
> **4. Why FGSM and will using PGD change the performance?**
>
> As we stated in Sec. 3.4, Alg. 2, and appendix A.1 crafting attacks for DM requires carefulness. Note that we implemented **FGSM with random start**--Eq.7. FGSM was chosen because it was the simpler and faster attack, since training with PGD is too computationally expensive for large datasets. Secondly, papers like [A] suggest that the addition of random start could make it resistant to multi-step attacks (PGD).
> Nevertheless, following this remark, we will mention PGD-training as future work in Sec. 5.
>
>
> **5. Discussion and References on Adv. Purification and Denoising for AT.**
>
>  Following your remark and also rev. `a6SF`, we will add a minisection at the end of Section 4. We will cite and discuss the suggested paper [1]. Please see the response to reviewer `a6SF`.
>
> Remaining questions:
>
> >  if diffusion models are successfully attacked, where the generated images are all wrongly classified, will the FID and IS be influenced?
>
> In the context of DMs, in this case considered to be **unconditional**, a successful attack disrupts model generation (it generates nonsensical images **see point 1**). Some samples are provided in Fig. 7 bottom. To reply to the question: if DMs are successfully attacked, they generate noise-like images, thereby disrupting both FID/IS and also classification. The case where DMs generate images to fool a classifier and FID and IS stay the same is not considered in our work but could be future work using ours as a purifier.
>
> > still not convinced that the robustness can be evaluated by FID and IS when defending against adversarial attacks
>
> Please, see the reply above in **point 1.**
>
> [A] Fast is better than free: Revisiting adversarial training, ICLR 2020

---

> > ### Comment · Reviewer_riys · 2025-04-09
> >
> > I thank the authors for their explanations. All my questions and concerns are addressed. I would like to increase my score by 1.

---

> > > ### Author Response · Authors · 2025-04-09
> > >
> > > We are sincerely grateful for your time and insightful feedback.
> > > We are pleased that the revisions contributed to improve the paper, and we greatly appreciate the increased score.

---

### Decision · Program_Chairs · 2025-05-01

**Decision:**

Reject

**Comment:**

This work studies adversarial training for diffusion models, highlighting its fundamental differences from adversarial training for classifiers. Overall, the paper is clearly structured and easy to follow. The conceptual logic is presented clearly, and the theoretical analysis provided is insightful and thorough. The major limitation is the insufficient experimental comparisons and evaluations. The authors provided some additional experiments in other datasets rather than CIFAR-10 during rebuttal, which is helpful.  However, the empirical evaluation and comparisons are still not sufficiently convincing. More large scale datasets, more strong attack methods, and other adversarial robustness methods are highly expected for comprehensive evaluation.